# Programmed Non-Apoptotic Cell Death in Hereditary Retinal Degeneration: Crosstalk between cGMP-Dependent Pathways and PARthanatos?

**DOI:** 10.3390/ijms221910567

**Published:** 2021-09-29

**Authors:** Jie Yan, Yiyi Chen, Yu Zhu, François Paquet-Durand

**Affiliations:** Cell Death Mechanism Group, Institute for Ophthalmic Research, University of Tübingen, Elfriede-Aulhorn-Strasse 7, 72076 Tübingen, Germany; jieyan19910809@hotmail.com (J.Y.); imyiyichen@gmail.com (Y.C.); esmezhu@outlook.com (Y.Z.)

**Keywords:** PKG, cGK, cGMP, photoreceptor, phototransduction

## Abstract

Programmed cell death (PCD) is a highly regulated process that results in the orderly destruction of a cell. Many different forms of PCD may be distinguished, including apoptosis, PARthanatos, and cGMP-dependent cell death. Misregulation of PCD mechanisms may be the underlying cause of neurodegenerative diseases of the retina, including hereditary retinal degeneration (RD). RD relates to a group of diseases that affect photoreceptors and that are triggered by gene mutations that are often well known nowadays. Nevertheless, the cellular mechanisms of PCD triggered by disease-causing mutations are still poorly understood, and RD is mostly still untreatable. While investigations into the neurodegenerative mechanisms of RD have focused on apoptosis in the past two decades, recent evidence suggests a predominance of non-apoptotic processes as causative mechanisms. Research into these mechanisms carries the hope that the knowledge created can eventually be used to design targeted treatments to prevent photoreceptor loss. Hence, in this review, we summarize studies on PCD in RD, including on apoptosis, PARthanatos, and cGMP-dependent cell death. Then, we focus on a possible interplay between these mechanisms, covering cGMP-signaling targets, overactivation of poly(ADP-ribose)polymerase (PARP), energy depletion, Ca^2+^-permeable channels, and Ca^2+^-dependent proteases. Finally, an outlook is given into how specific features of cGMP-signaling and PARthanatos may be targeted by therapeutic interventions.

## 1. Introduction

In multicellular organisms, tight regulation and control of cell death are essential for development, tissue homeostasis, and survival [1]. Different cell types, tissues, and varying physiological conditions have resulted in the evolution of a multitude of cell death mechanisms for which collectively the term “programmed cell death” (PCD) has been introduced [2]. PCD mechanisms, such as apoptosis, usually involve a genetically controlled, program-driven activation of biochemical processes and cellular machinery that result in systematic cellular self-destruction [3].

Hereditary retinal degeneration (RD) is a genetically diverse group of diseases that typically result in progressive photoreceptor cell death, severe visual handicap, and blindness [4]. The most common disease within the RD group is retinitis pigmentosa (RP) [5], in which patients initially experience night blindness and gradual constriction of the visual field until complete blindness sets in [6,7]. The disease typically displays a two-step progression where primary loss of rod photoreceptors is followed by secondary degeneration of cone photoreceptors [6,7]. Approximately one in four thousand people are affected by RP [5,6]. Overall, RD-type blinding diseases are considered to be untreatable [8]. Key questions in the field of RD research concern the cellular mechanisms that bring about photoreceptor loss and whether these may be targeted by therapeutic interventions to prevent or delay the progression of RD.

While over the last 2–3 decades, research has overwhelmingly focused on apoptosis as a causative mechanism for RD, recent evidence indicates that apoptosis only plays a minor role in photoreceptor degeneration [9,10,11,12]. Instead, the importance of non-apoptotic mechanisms, for instance, triggered by exceedingly high levels of cyclic-guanosine-mono-phosphate (cGMP), is increasingly recognized [13]. Hence, for the purposes of this review, we will mention apoptosis only briefly to then focus on non-apoptotic cell death mechanisms. This will include cGMP-dependent cell death [13] and PARthanatos [14] as well as enzymes involved in either or both degenerative pathways. Finally, an outlook is given into how future therapeutic approaches may use an improved mechanistic understanding of PARthanatos and cGMP-signaling.

## 2. Photoreceptor Physiology and Phototransduction

Photoreceptors mediate the conversion of a photon of light into an electrochemical message that can be interpreted by second-order neurons, a process referred to as phototransduction. A feature of vertebrate phototransduction is their separation into rod and cone photoreceptors, adapted, respectively, for night and day vision. While rods and cones utilize distinct isoforms of the protein components of the transduction cascade, in both types of photoreceptors, cGMP plays a key role [15]. In darkness, cGMP concentration is controlled by the dynamic equilibrium of its synthesis and hydrolysis governed by the basal activities of retinal membrane guanylyl cyclases (GCs) and phosphodiesterase (PDE6), respectively [16,17]. Cyclic nucleotide-gated ion channels (CNGCs), belonging to the superfamily of pore-loop cation channels, are ion channels that are located in the outer segment plasma membrane of photoreceptors and that are activated by the binding of cGMP or cAMP [18,19]. Interestingly, mutations in genes encoding for CNGC subunits can cause retinal degeneration [18,20], attesting to their importance for photoreceptor physiology. A high level of free cGMP in darkness maintains ∼3% of CNGCs in the open state, allowing for an influx of Na^+^ and Ca^2+^ into the outer segment [16,17]. While Ca^2+^ is extruded from the outer segment by the Na^+^/Ca^2+^-K^+^ (NCKX) exchanger, Na^+^ is pumped out of the inner segment by the adenosine-5′-triphosphate (ATP) -driven Na^+^/K^+^ exchanger (NKX). This continuous flow of Na^+^ ions forms the circulating dark current (Figure 1) [21].

As opposed to other neuronal cell types, photoreceptors are relatively depolarized in their resting state, which in turn leads to a constant synaptic release of the neurotransmitter glutamate. Photon absorption by an opsin protein in the outer segment leads to a conformational change, allowing the opsin to bind and activate the G-protein transducin, which then binds the inhibitory γ-subunit of phosphodiesterase-6 (PDE6), releasing the catalytic PDE6αβ subunits to hydrolyze cGMP. The subsequent decline of cGMP concentrations leads to the closure of CNGCs and the reduction of the dark current. The net decrease in the influx of positively charged Na^+^ and Ca^2+^ leads to a hyperpolarization of the photoreceptor cell and the cessation of synaptic glutamate release [22,23].

Low intracellular Ca^2+^ activates guanylyl cyclase-activating protein (GCAP), stimulating GC, and increasing cGMP synthesis [24,25,26]. Rising cGMP, in turn, re-opens CNGC and facilitates the recovery of the light response [27,28]. As we will see in the next chapter, the very high energy consumption of photoreceptor phototransduction has ramifications for cell death mechanisms.

## 3. Retinal Energy Metabolism and Cell Death

The retina is one of the most energy-demanding tissues in the body [29], with a large part of its ATP consumption caused by NKX activity and maintenance of the dark current [30,31]. Remarkably, the retina relies mostly on the relatively energy-inefficient glycolysis even in the presence of oxygen, rather than using oxidative phosphorylation, which could provide 18 times as much ATP per mol of glucose [32]. This phenomenon was recognized already by Otto Warburg in the early 1920s and is referred to as “aerobic glycolysis” or as the “Warburg effect” [33]. While the significance of aerobic glycolysis in the retina is, in part, still unclear, several studies have proposed that it may enhance the anabolic activity of photoreceptors [34,35,36].

At any rate, the availability of ATP will determine what kind of cell death mechanism can be operated by a cell. For instance, apoptosis as an active, ATP-dependent process cannot be executed without sufficient ATP, while necrotic forms of cell death require much lower ATP-levels or no ATP at all for their execution [37,38,39,40]. Several different pathological situations may entrain a depletion of ATP. For instance, a cGMP-mediated overactivation of CNG channels may lead to cytosolic Ca^2+^ overload. High cytosolic Ca^2+^, together with other causal factors, may lead to an increase in mitochondrial membrane permeability and subsequent membrane depolarization, increased mitochondrial reactive oxygen species (mROS) generation, cytochrome C release, and apoptosis [37,38,41]. In turn, ROS production may deteriorate mitochondrial function even further [42,43]. On one hand, this will decrease photoreceptor ATP production, while on the other hand, ATP consumption will be increased by the extrusion of Ca^2+^ from the cell soma and synapse via the ATP-driven plasma membrane Ca^2+^-ATPase (PMCA) [30]. The net effect would be a complete depletion of ATP and then cell death via non-apoptotic, ATP-independent mechanisms.

ATP is also intricately linked to nicotinamide adenine dinucleotide (NAD^+^), a metabolite that serves as a cofactor for hydrogen transfer. As such NAD^+^ is vital for the operation of glycolysis and ATP-synthesis in the mitochondria [44]. Thus, if NAD^+^ levels fall below a critical threshold, key metabolic processes capable of delivering ATP will cease to function. Such a situation may occur in PARthanatos, where the activity of poly(ADP-ribose)polymerase (PARP) may consume excessive amounts of NAD^+^, thereby indirectly also depleting intracellular ATP [45]. In RD-type diseases, the depletion of ATP consumption and NAD^+^ may be a significant concern for disease pathogenesis.

## 4. Apoptosis

Apoptosis is a form of PCD that occurs not only during development or aging, but also as a defense system. Apoptosis can be activated by intrinsic and extrinsic signaling [46]. The intrinsic pathway, also known as the mitochondrial pathway, is driven by a signal within a cell, inducing the expression of proapoptotic genes and proteins, including those belonging to the BCL-2 family, which form the mitochondrial outer membrane permeabilization (MOMP), allowing for the release of cytochrome C to the cytoplasm where it binds to apoptotic protease-activating factor-1 (APAF1). The resultant multimeric cytochrome-c/APAF1 complex activates caspase-9 [47], which in turn cleaves and activates downstream executioner caspases, such as caspase-3 and -7. The extrinsic pathway is triggered by the activation of cell-surface death receptors, such as the tumor necrosis factor family. Later, caspase-8 gets activated, which may then cleave and activate downstream caspases-3 and -7 directly, or, alternatively, may activate BCL-2 family proteins, executing the same steps as in classical apoptosis [48,49,50].

In the past, apoptosis was regarded as a primary degenerative mechanism in RD [51], yet, in the last decade an increasing amount of evidence pointed to the importance of non-apoptotic mechanisms, including cell death pathways triggered by high cGMP [12,13]. One of the confounding factors that has made it difficult to separate apoptotic PCD from the cell death mechanisms that actually underlie RD is the fact that apoptosis is a prominent feature of retinal development [52] and that many commonly used RD animal models display mutation-induced photoreceptor cell death in the very same time frame [38].

## 5. PARP Activity and PARthanatos

PARP-type enzymes catalyze the transfer of ADP-ribose to target proteins [53] and can sequentially add ADP-ribose units from NAD^+^ to form polymeric ADP-ribose chains (PAR) [54]. There are at least 18 PARP family members encoded by different genes and with a shared homology in the conserved catalytic domain [55]. While PARP activity was originally associated with DNA repair enzymes and gene regulation [56,57], it may also be the primary driver for a specific form of cell death, termed PARthanatos [14]. This relatively recently discovered non-apoptotic cell death process is characterized by PARP overactivation, accumulation of PAR, and nuclear translocation of apoptosis-inducing factor (AIF) from the mitochondria. As such, PARthanatos is likely involved in various retinal degenerative diseases [58,59].

### 5.1. The Core of PARthanatos: PARP and PAR Polymers

PARP-1 is probably a central mediator of this cell death mechanism since the majority (>90%) of PAR polymer synthesis typically stems from PARP-1 [60,61]. PARP-1 can mediate cell death when high levels of DNA damage activate PARP-1 to a degree that depletes cellular NAD^+^ levels. The subsequent depletion of ATP decreases all energy-dependent functions and leads to cell death before DNA repair can be accomplished [62].

Poly(ADP-ribosyl)ation (PARylation) was first described by Chambon and colleagues more than 50 years ago [63]. For PAR chains formation, NAD^+^ molecules must be cleaved by PARP, and resulting ADP-ribosyl units must be attached to already existing ones [64,65]. The DNA break-induced activation of PARP-1 triggers PARylation of proteins, including PARP-1 itself (auto-PARylation), to facilitate the recruitment of DNA repairing enzymes that contribute to DNA repair near the DNA damage site. Although PARylation is primarily a survival mechanism, high PARylation activity can cause regulated cell death in cells with excessive DNA damage [60]. Suppressing PARylation rescues cells from PARthanatos, attesting to the important role PARylation plays in PARP-1 and DNA damage-induced cell death [66]. Remarkably, PAR itself exhibits dose-dependent toxicity in neurons when exogenously administered via BioPorter, a cationic lipid formulation that facilitates PAR entry into cells [67].

In physiological conditions, several mechanisms counterbalance excessive PARP-1 activity. (1) PAR molecules are rapidly catabolized by poly-(ADP-ribose) glycohydrolase (PARG) and ADP-ribosyl protein lyase [68,69]; (2) auto-PARylation of PARP-1 downregulates its activity by interfering with interactions between the DNA and DNA binding domain [70]; (3) accumulation of nicotinamide as a by-product of NAD^+^ consumption inhibits PARP-1 and may act as a negative feedback signal [45,62].

### 5.2. PAR-Dependent Translocation of AIF

AIF was originally identified as a soluble 57-kDa fragment, which upon dissipation of the mitochondrial membrane potential is released from the mitochondria and translocates to the nucleus in a caspase-independent manner [71]. PAR induces a conformation change in AIF that lowers its affinity to the mitochondrial outer membrane leading to its release [72]. Hence, after PARP-1 overactivation, AIF can be released into the cytoplasm and translocate further to the nucleus, promoting cell death [73]. The protective effect by blocking mitochondrial AIF release or reducing AIF abundance indicates that AIF plays a crucial role in PARthanatos [65,74,75]. Furthermore, AIF can recruit macrophage migration inhibitory factor (MIF) to the nucleus, where MIF cleaves genomic DNA into large-scale fragments via its nuclease activity [73]. Such DNA damage may lead to even more PARP-1 activation, effectively forming a feedback loop that accelerates NAD^+^ depletion, mitochondrial dysfunction, and DNA degradation.

### 5.3. Crosstalk between PARthanatos, Ca^2+^, and Calpain-Type Proteases

As described above (Section 3), ATP depletion caused by excessive PARP-1 activity [76] will compromise a cell’s capability for Ca^2+^ extrusion via ATP-dependent PMCA (Figure 1). The resulting rise in intracellular Ca^2+^ may lead to Ca^2+^-dependent activation of calpain-type proteases [48,53]. While calpain may be involved in AIF release from mitochondria [77], during PARthanatos, AIF release can also happen independently of calpain activity [78]. Still, other than via calpain activation, high intracellular Ca^2+^ levels, and Ca^2+^ sequestration into mitochondria can also lead to the generation of reactive oxygen species and mitochondrial dysfunction, further promoting the execution of PARthanatos [79,80].

## 6. cGMP-Dependent Cell Death in RD

Many genetically distinct forms of RD display a substantial increase in the intracellular photoreceptor concentration of cyclic guanosine monophosphate (cGMP) [12,81,82]. We have previously proposed for this to constitute a new cGMP-dependent cell death pathway for photoreceptor degeneration [48] that would apply to any disease-causing mutation that raises intracellular cGMP levels. In this mechanism, overactivation of the prototypic cGMP targets CNGC and protein kinase G (PKG), produces excessive Ca^2+^ influx and protein phosphorylation, respectively [12]. Both CNGC and PKG alone or in concert precipitate cell death: CNGC-mediated Na^+^ and Ca^2+^ influx may strain the energy metabolism and activate calpain [83,84]. PKG-dependent phosphorylation is associated with histone deacetylase (HDAC) activation [85], which in turn appears to be connected to the activation of PARP [86].

### 6.1. RD Mutations Associated with High Photoreceptor cGMP

Regarding the genetically very heterogenous group of RD-type diseases, a relevant question concerns the generality of cGMP-dependent cell death. Indeed, excessive accumulation of cGMP in photoreceptors has been observed in various genetically distinct RD mutants [12,81,82,87,88], suggesting cGMP as a near-universal trigger for non-apoptotic PCD mechanisms in RD. Disease-causing mutations may, for instance, result in gain-of-function in genes involved in cGMP synthesis [89,90]. Similarly, loss of function in genes downregulating cGMP can cause RD [91,92,93], and so can mutations in cGMP-signaling targets [19,84,94], such as in CNGC genes. For further details on mutations in RD disease genes that have been connected to exceedingly high levels of cGMP in photoreceptors, please refer to [47,95].

### 6.2. cGMP-Gated Ion Channels, Ca^2+^ -Influx, and Cell Death

As mentioned in Section 2, cGMP-activation of CNGCs plays a central role in phototransduction [19]. These channels are heterotetramers consisting of CNGA1 and CNGB1 subunits in rod photoreceptors and CNGA3 and CNGB3 subunits in cones [19]. Elevated photoreceptor cGMP levels may overactivate CNGC and increase Na^+^ and Ca^2+^ influx. Knockout of the *Cngb1* gene caused rod CNGC dysfunction and significantly delayed photoreceptor degeneration in *Pde6b* mutant mice [84,87]. On the other hand, loss-of-function mutations in CNGC genes cause RD [95], and even further pharmacological inhibition of CNGC in *Pde6b* mutant retina accelerated photoreceptor cell death [96]. A possible explanation for these seemingly contradictory outcomes could be that a low concentration of intracellular Ca^2+^ activates GCAPs to stimulate GCs and increase cGMP synthesis [24,25,26]. In *Cngb1^-/-^* mice, the introduction of an additional knockout of the PKG1 gene (i.e., *Prkg1^-/-^*) delays photoreceptor degeneration [87], suggesting that photoreceptor death ultimately is mediated by PKG-dependent processes.

The activity of CNGCs depolarizes photoreceptors to the extent that voltage gated Ca^2+^ channels (VGCC) in the soma and synapse open [97]. This results in further Ca^2+^ influx that must be extruded by PMCA at the expense of additional ATP and which, otherwise, could lead to the activation of calpain-type proteases, which may promote cell destruction [98,99,100]. Accordingly, the idea that high Ca^2+^ may be responsible for cell death [48], and by extension for photoreceptor degeneration [101], has motivated several studies attempting to block either CNGC or VGCC for therapeutic purposes. Unfortunately, these attempts generally failed to yield tangible success [102,103,104], which may indicate that Ca^2+^ influx is, in fact, not as relevant for photoreceptor cell death as previously thought [96]. The lack of therapeutic effect of Ca^2+^ channel blockers also suggests that the main drivers of cGMP-dependent photoreceptor cell death are mainly independent of Ca^2+^ [21].

### 6.3. Protein Kinase G: A Link between cGMP-Signaling and Cell Death?

PKG is a major downstream effector of cGMP-signaling pathways [105]. Conventionally, cGMP-PKG signaling is often seen as protective, especially in a neuronal cell context [106,107,108]. However, overactivation of PKG can also cause cell death [109,110,111], and in the retina and in photoreceptors, a link between excessive cGMP-signaling, PKG activity, and cell death has been well established [81,112,113]. Moreover, application of PKG inhibitors in vivo in RD retina resulted in photoreceptor protection [81,114], and abolishing *Prkg1* expression in the mouse promoted rod photoreceptor survival [87]. Conversely, knockout of the *Prkg2* gene in the mouse protected cone photoreceptors in a model for hereditary cone degeneration [115]. However, it is worth noting that the exact PKG expression patterns in the retina are still partially unresolved and that in situ hybridization [116,117] and immunostaining studies [118] are not in entire agreement with each other as to which PKG isoform is expressed in which retinal cell type.

Another open question is the nature of the protein targets that PKG may phosphorylate in the retina and how such phosphorylation could bring about photoreceptor cell death. A recent study used protein phosphorylation array technology to identify PKG targets in retinal tissue lysates. While some of the previously known PKG targets—such as vasodilation-stimulated phosphoprotein (VASP) and cyclic AMP response element-binding (CREB)—were confirmed, many novel substrates were also found, including ryanodine receptor-1 (RYR1) and 6-phosphofructo-2-kinase/fructose-2,6-biphosphatase 3 (PFKFB3) [119]. The exact significance of these and other PKG targets for photoreceptor degeneration will have to be elucidated in future studies.

### 6.4. Histone Deacetylase Activity as an Event Downstream of PKG

Modification of histones by the activity of histone deacetylases (HDACs) plays a key role in epigenetic regulation of gene expression by changing the structure of chromatin and by modulating the accessibility for transcription factors to their target DNA sequences [120]. Notably, excessive activation of HDACs in photoreceptors has been observed in connection with the accumulation of cGMP and activation of PKG [12,86,121], and inhibition of HDAC protects the retina from cGMP-induced neurodegeneration in several RD mutants [122,123,124,125,126,127,128]. Nonetheless, the mechanism of HDAC activation remains mysterious. While in *C. elegans*, PKG-dependent phosphorylation appears to activate HDACs [85], it is unclear whether such direct PKG-HDAC interactions also happen in higher vertebrates, even though HDAC1 can serve as a substrate for PKG in vitro [129]. Overall, HDAC activity may have both positive and negative effects on cell survival, and a PKG-dependent disturbance in the equilibrium of different HDAC functions—whether direct or indirect—could play an important role in neurodegeneration.

### 6.5. PARP: A Link between PARthanatos and cGMP-Dependent Cell Death

As discussed above, PARP-1 is the central mediator of PARthanatos due to its ability to synthesize PAR polymers [60,61]. In addition, PARP activity is strongly increased in photoreceptors during the progression of RD [12,130,131]. Remarkably, the inhibition of HDAC activity with trichostatin A led to a decrease in photoreceptor PAR accumulation, indicating that PARP activity may occur downstream of HDAC [86,125]. This may be linked to HDAC-mediated removal of acetylated residues from histones, leading to chromatin condensation and transcriptional repression. Chromatin condensation, in turn, impairs the recruitment of DNA repair factors and results in the accumulation of DNA breaks [132]. Due to continuous exposure to endogenous and exogenous DNA-damaging insults, cells accumulate DNA damage such as single-strand DNA breaks (SSBs) and double-strand DNA breaks (DSBs). This requires constant surveillance and activation of the DNA repair response [132] that is facilitated by PARP-1 [52]. Paradoxically, inhibition of PARP significantly delays photoreceptor loss in cGMP-dependent cell death [133,134,135,136], and PARP-1 gene knockout increases resistance to RD [137], indicating that PARP activity, perhaps through its consumption of NAD^+^, is an important driver of photoreceptor degeneration.

Taken together, PARP activity may link PARthanatos (Section 5) on the one hand with cGMP-dependent photoreceptor degeneration [48] and on the other hand with photoreceptor energy metabolism (Section 3).

## 7. Therapy Developments Targeting Programmed Cell Death

The degenerative photoreceptor pathways discussed thus far may provide targets for the rational design of therapeutic interventions that could prevent or slow down the progression of RD. Numerous studies have attempted to exploit mechanistic insights for therapy development purposes, and many of these works have focused on apoptosis as the presumed causative mechanism (Table 1).

### 7.1. Apoptosis as a Target for Therapeutic Intervention in RD

RD was initially thought to be driven by apoptosis [51], motivating many interventional studies that aimed to block different steps of the apoptotic cascade [138,139,140]. Unfortunately, these approaches were mostly ineffective, even though virtually the whole apoptotic cascade has been targeted. For instance, neither the pharmacological inhibition of caspase-type proteases [139], nor the genetic manipulation of Bcl-2 [141], Bcl-XL [142], c-fos [146], caspase-3 [140], or caspase-7 [147] promoted long-term photoreceptor survival. On the other hand, proapoptotic protein Bcl-2-associated X protein (BAX) activation was observed in three different animal models for RD [148]. However, the role of BAX in RD also seems ambiguous as its gene knockout may delay cell death of rod photoreceptors but not that of cones in Rpe65 KO animals [138]. Together, most of the available evidence does not suggest a major role for apoptosis in RD and hence makes it seem unlikely that therapeutic interventions targeting apoptosis can be successful.

### 7.2. Targeting PARthanatos and PARP Activity

PARthanatos occurs in a highly choreographed, multistep fashion, and several steps in the cascade could serve as therapeutic targets for managing diseases associated with cell death [143]. Since PARthanatos is characterized by overactivation of PARP-1 [60,61], PARP inhibitors should prioritize saving photoreceptor degeneration. Currently, PARP inhibitors are mostly used for cancer therapy due to their ability to prevent DNA repair, and several PARP inhibitors are being tested clinically or have already been approved for clinical use [144]. Thus, it could be economically efficient to repurpose PARP inhibitors for applications beyond oncology, including in RD [136].

Olaparib became the first PARP inhibitor to be approved by the FDA to treat metastatic breast cancer in January 2018 [149]. In the *Pde6b* mutant *rd1* mouse, Olaparib significantly delayed photoreceptor loss [133] and affected the release of extracellular vesicles associated with immune modulation, tumor invasion, regeneration, degenerative processes, cellular communication, cell homeostasis, and neovascularization [134]. In addition, other PARP inhibitors also showed its neuroprotective effects in the retina. For instance, BMN-673, 3-aminobenzamide [136], and PJ34 [145] decreased PARylation and reduced photoreceptor cell death. While these data confirm PARP in general as a therapeutic target for RD, it is still uncertain whether PARP-1 is the main target or other PARP isoforms also need to be considered. Moreover, a better understanding of how PARP activity precipitates cell death will likely be needed to develop a truly effective therapy.

### 7.3. Novel Therapeutic Approaches Targeting cGMP-Dependent Cell Death

In a large subset of RD patients, cGMP-dependent cell death may be the prevalent pathogenic mechanism, highlighting this pathway for gene- and mutation-independent therapeutic purposes [12,48]. The most likely targets for high cGMP levels in the photoreceptor are PKG and/or CNGC (Figure 1) [87]. cGMP-induced overactivation of CNGC, presumably causing a dangerous inflow of Ca^2+^, has been proposed as a key event in photoreceptor degeneration [91,113], but as we discussed above, singly inhibiting Ca^2+^ might not rescue photoreceptors [96,103,104,150], due to feedback of low-Ca^2+^ levels increasing cGMP [24,25,26]. On the other hand, PKG appears as an essential mediator of cGMP-dependent cell death [81,112]. Accordingly, an inhibitory cGMP analogue delivered via a nanosized liposomal carrier to facilitate transport across the blood-retinal barrier was highly effective at protecting photoreceptors in three different in vivo RD models [114].

Still, interfering with PKG activity would not per se reduce the high intracellular cGMP levels caused by RD mutations. Thus, an alternative approach could be to target the cGMP synthesis pathway. Here, the enzyme inosine monophosphate dehydrogenase (IMPDH), which catalyzes the rate-limiting step of GTP production [151] stands out as an exciting novel target. Yang et al. [82] demonstrated that mycophenolate mofetil, a prodrug of mycophenolic acid that reversibly inhibits IMPDH [152], reduced photoreceptor cGMP and preserved photoreceptors in Pde6b mouse mutants.

Altogether, these data make cGMP-signaling appear as a lovely pathway for RD therapy development. As a note of caution, it is crucial to consider that cGMP-signaling covers essential functions in many cells of the body. Thus, a prospective drug will need to be either highly specific for its target in photoreceptors or delivered with a system that supplies the drug specifically to the retina and the photoreceptors.

## 8. The Future Therapeutic Research in RD

Although many novel therapies have been discovered (Section 7), there are still challenges in clinical translation. For instance, RD-type diseases show an enormous genetic heterogeneity with disease-causing mutations in more than 270 genes [48]. Since each of these disease genes can carry from several dozens to several hundred or more individual mutations [153,154], we may, at present, estimate the total number of disease mutations to amount to several tens of thousands. This situation severely hinders the design of clinical trials as the numbers of patients carrying a specific disease-causing mutation will be small, even in a best-case scenario. However, a careful choice of the patients to be included in a clinical trial with precisely known genotypes is critical for success during clinical testing.

Another problem for clinical translation is a lack of in vivo biomarkers due to the incomplete understanding of the underlying photoreceptor cell death mechanisms. Ideally, the biomarker could be used for the rapid assessment of treatment efficacy and should be allowed for live non-invasive visualization of cell death in the retina. Recent developments in magnetic resonance imaging (MRI) suggest that it may be possible to non-invasively observe oxidative stress or the production of free radicals in the retina of animals [155]. In addition, cGMP, a potential blood-based parameter, is increased in blood from RP patients compared to healthy counterparts [156,157]. This may be related to the excessive cGMP levels in the photoreceptors of many RP types, i.e., the phenomenon discussed in several sections above, especially since at least some of the patients had mutations in the *PDE6A* gene [157].

## 9. Concluding Remarks

The question as to what type of PCD mechanism governs cell death in RD-type diseases has been investigated for at least three decades, and many findings have been contentious for many years. It is thus worth noting that this review is neither exhaustive nor unbiased. While the recent decade has yielded a large body of evidence pointing at cGMP-dependent cell death and PARthanatos as likely causative mechanisms in RD, future research will have to confirm this, hopefully delivering effective RD therapies to patients.

## Figures and Tables

**Figure 1 ijms-22-10567-f001:**
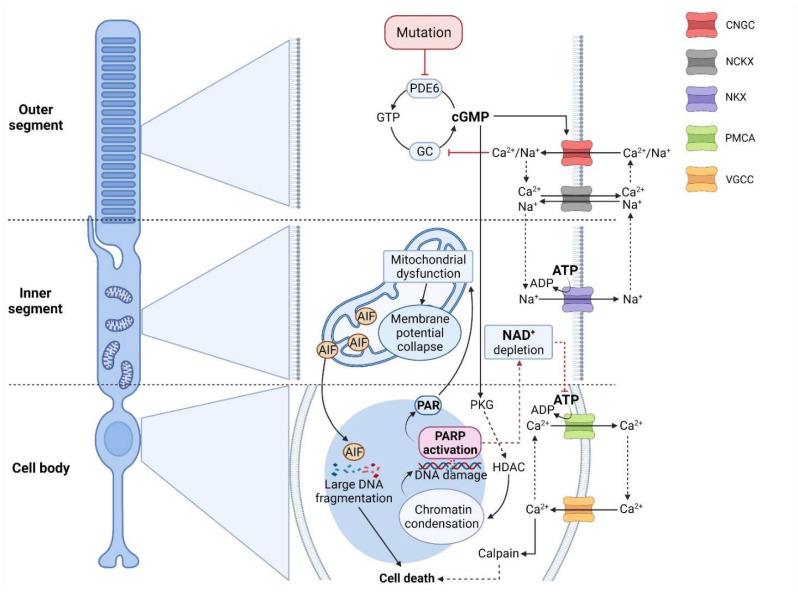
Crosstalk between cGMP-signaling and PARthanatos in different photoreceptor compartments. In RD-type diseases, photoreceptor degeneration is often initiated by high cGMP levels, caused, for instance, by mutations affecting phosphodiesterase-6 (PDE6). On the one hand, cGMP activates protein kinase G (PKG), which is associated with histone-deacetylase (HDAC), leading to chromatin condensation and DNA damage. This, in turn, may trigger the over-activation of histone deacetylase (HDAC) and indirectly PARP, leading to NAD^+^/ATP depletion. In addition, PAR polymers produced by PARP can cause mitochondrial dysfunction and translocation of apoptosis-inducing factor (AIF) to the nucleus. On the other hand, cGMP opens the cyclic-nucleotide-gated channel (CNGC), promoting Na^+^/Ca^2+^ influx and voltage change in the outer segment. In turn, Ca^2+^ inhibits guanylyl cyclase (GC), which limits cGMP synthesis from GTP under physiological conditions. The CNGC-dependent voltage change may further open voltage-gated-Ca^2+^ channels (VGCC) in soma and synapse, leading to more Ca^2+^ influx. PARP-dependent ATP-depletion may reduce NKX-mediated ion extrusion, resulting in higher intracellular Ca^2+^ levels. These, in turn, may be linked to calpain-type protease activation, precipitating cell death.

**Table 1 ijms-22-10567-t001:** An overview of current therapeutical targets in three different PCDs.

Cell Death Mechanism	Targets	Methods	Results	References
Apoptosis	Caspase-type proteases	Pharmacological inhibition	No effect/minor delay of photoreceptor loss	[136]
	Bcl-2, Bcl-XL, c-fos, caspase-3, caspase-7	Gene knockout		[137,138,139,140,141]
	BAX	Gene knockout	Only saving rods	[135,142]
PARthanatos	PARPs	Pharmacological inhibition	Delayed photoreceptor loss	[130,133,143]
cGMP-dependentcell death	CNGC	Gene knockout, pharmacological inhibition	Photoreceptor protection with gene knockout, no protection after pharmacological inhibition	[95,144]
	VGCC	Gene knockout, pharmacological inhibition	Minor delay after gene knockout, no protection pharmacological inhibition	[102,103]
	PKG	Pharmacological inhibition	Morphological and functional photoreceptor protection	[111]
	IMPDH	Pharmacological inhibition	Reduced photoreceptor cGMP, photoreceptor protection	[81,145]

## Data Availability

Not applicable.

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
