# Peer review of "Programmed Non-Apoptotic Cell Death in Hereditary Retinal Degeneration: Crosstalk between cGMP-Dependent Pathways and PARthanatos?"

_ijms, 2021, doi:10.3390/ijms221910567_

Round 1
Reviewer 1 Report
Yan et al. wrote a very interesting review describing the role of “Programmed non-apoptotic cell death in hereditary retinal degeneration: Crosstalk between cGMP-dependent pathways and PARthanatos?”. The manuscript represents an interesting way to discover new scenarios for retinal degenerations. I suggest only several minor revisions needed to update and improve the reliability of the paper:
- The “Chapter 2” lacks of most recent literature about ion channels and their involvement into retinal degenerations. Regarding these, I suggest to add the following references to manuscript PMID: 33374679 and PMID: 33801777.
- In the “Chapter 3”, the authors should link better the retinal energetic metabolism with retinal physiopathology, also depicting the role of oxidative stress. Regarding these, I suggest to add the following references to manuscript PMID:34058230 and PMID: 34440511.
- Finally, manuscript requires English revisions and typos correction.
Author Response
- The “Chapter 2” lacks of most recent literature about ion channels and their involvement into retinal degenerations. Regarding these, I suggest to add the following references to manuscript PMID: 33374679 and PMID: 33801777.
Answer: Thank you for your advice. We have now included the said references (PMID: 33374679) in the text line 69-71. However, in Chapter 2, we are only talking about the photoreceptor physiology rather than retinal pathophysiology, because PARthanatos and cGMP-dependent cell death may be associated with Ca2+ and energy depletion. It is a general background introduction for the novices. Thus, we did not include PMID: 33801777.
- In the “Chapter 3”, the authors should link better the retinal energetic metabolism with retinal physiopathology, also depicting the role of oxidative stress. Regarding these, I suggest to add the following references to manuscript PMID:34058230 and PMID: 34440511.
Answer: This is another good idea; however, we note that a causal link between oxidative stress and photoreceptor death has not been clearly established in hereditary retinal degeneration. Nevertheless, to allude to such possibility we have now added extra sentences in chapter 3 (line 106 - 111) to describe the retinal energetic metabolism with retinal physiopathology, and oxidative stress.
- Finally, manuscript requires English revisions and typos correction.
Answer: Thank you for your suggestion. We have done that thoroughly.
Reviewer 2 Report
Dear Authors,
Manuscript ID: ijms-1385335 entitled, "Programmed non-apoptotic cell death in hereditary retinal degeneration: Crosstalk between cGMP-dependent pathways and PARthanatos?" is a compelling narrative review that, in my opinion, will influence researchers to reconsider and relook their perspectives. In writing this review, the authors intended to study the non-apoptotic programmed cell death (PCD) that points to PARthanatos and cGMP-dependent cell death as a plausible mechanism for retinal degeneration. The authors' intended conclusion defined cGMP signaling as an alternative pathway for therapy development, as expected. Even though the references in the manuscript are somewhat dated, the information they provide is still valid today and is adequate. The manuscript is written clearly and concisely, with a simple narrative structure. Yet, very few of my suggestions should be considered for inclusion in the final version.
-
Would you mind creating an illustration that briefly covers Section 7?
-
In section 7.3, line 373, signaling is covers……….. remove "is"
-
Please include the "future prospective" section briefing a few possible/hypothetical targets in cGMP signaling that is best for RD therapy development.
Author Response
- Would you mind creating an illustration that briefly covers Section 7?
Answer: Thank you for your advice. We have added a table (Table 1) above Section 7 to summarize the therapies and outcomes for different cell death mechanisms.
- In section 7.3, line 373, signaling is covers……….. remove "is"
Answer: Thank you, we have revised it.
- Please include the "future prospective" section briefing a few possible/hypothetical targets in cGMP signaling that is best for RD therapy development.
Answer: Thank you, the possible targets in cGMP signaling for RD therapy development have been discussed in chapter 7.2 and 7.3. However, we wrote a new future prospective chapter (chapter 8) to talk about the difficulties of clinical translation and the possibility to develop biomarkers based on cGMP-signaling.